# Colorectal Air–Liquid Interface Organoids Preserve Tumour-Immune Architecture and Reveal Local Treg Expansion After PD-1 Blockade

**DOI:** 10.3390/cancers18010132

**Published:** 2025-12-30

**Authors:** Laura Córdoba, Francisco J. Cueto, Ramón Cantero-Cid, Rebeca Abad-Moret, Esteban Díaz, Jaime Álvarez-Benayas, Jesús Fernández-Felipe, Jesús Jiménez-Rodríguez, Daniel Arvelo-Rosario, Pablo Mata-Martínez, Marina Arranz-Álvarez, Yaiza Pedroche-Just, Sandra Nieto-Torrero, Jaime Valentín-Quiroga, Verónica Terrón-Arcos, Jaime Fernández-Pascual, Paloma Gómez-Campelo, Nieves Cubo-Mateo, Olivia Fernández-Medina, Laura Hurtado-Navarro, Gonzalo Sáenz de Santa María, Julia del Prado-Montero, Agustín L. Santos, Roberto Lozano-Rodríguez, Carlos del Fresno, Eduardo López-Collazo

**Affiliations:** 1Biobank Platform, IdiPAZ, La Paz University Hospital, 28046 Madrid, Spain; 2The Innate Immune Response Group, IdiPAZ, La Paz University Hospital, 28046 Madrid, Spain; 3Tumour Immunology Lab, IdiPAZ, La Paz University Hospital, 28046 Madrid, Spain; 4Department of Medical Oncology, La Paz University Hospital, 28046 Madrid, Spain; 5Digestive Surgery Service, La Paz University Hospital, 28046 Madrid, Spain; 6Digestive Surgery Department, Faculty of Medicine, Universidad Autónoma de Madrid, 28046 Madrid, Spain; 7Translational Research and Innovation in Surgery Group, La Paz University Hospital, 28046 Madrid, Spain; 8ARIES Research Group, Universidad Antonio de Nebrija, 28248 Madrid, Spain; 9Faculty of Health Sciences, Universidad UNIE, 28015 Madrid, Spain; 10Immunomodulation Laboratory, IdiPAZ, La Paz University Hospital, 28046 Madrid, Spain; 11Biomedical Research Networking Center (CIBER), Respiratory Diseases (CibeRes), 28029 Madrid, Spain

**Keywords:** air–liquid interface organoids, Matrigel organoids, immune infiltrates, nivolumab, regulatory T cells

## Abstract

Understanding how colorectal tumours interact with immune cells is critical for improving immunotherapy responses. In this study, we used air–liquid interface organoids derived from patient tumours and adjacent healthy colon tissues to reproduce the original tumour microenvironment in the laboratory. These organoids preserved the three-dimensional tissue structure, the native immune infiltrates, and the genetic mutations of the original samples more faithfully than conventional Matrigel-based cultures. When compared with Matrigel organoids co-cultured with blood immune cells, the air–liquid interface model more accurately reflected the immune and molecular features of each patient’s tissue. Notably, treatment with the immune checkpoint inhibitor nivolumab (anti–PD-1) led to a consistent expansion of regulatory T cells, recapitulating a mechanism of adaptive resistance. This approach offers a physiologically relevant platform to study colorectal cancer–immune interactions and to predict patient-specific responses to immunotherapy.

## 1. Introduction

Colorectal cancer (CRC) is the third most common cancer worldwide, with an increasing incidence globally [1]. Although well-established clinical guidelines have contributed to improving survival rates through early diagnosis and enhanced treatment [2], many patients continue to face challenges, particularly with resistance to standard-of-care therapies. The advent of patient-derived organoids (PDOs) offers a promising avenue for the development of personalised medicine strategies.

Tumour organoids are three-dimensional (3D) structures with self-organizing and self-renewing properties, closely mimicking the structure, genetics and in vivo functions of their original tissue [3,4]. Numerous studies have highlighted the potential of CRC organoids in assessing individual patients’ responses to chemotherapeutic and agents [5,6], facilitating more personalised decision-making based on drug efficacy and toxicity profiles [7,8].

Traditionally, the generation of organoids has relied on the use of naturally derived extracellular matrixes, such as Matrigel [9,10]. However, despite its utility, this model faces significant challenges that limit its clinical application. One major limitation of Matrigel-based organoid models is that they consist exclusively of neoplastic epithelium, lacking key components of the tumour microenvironment (TME), such as infiltrating immune cells and stromal elements [11]. The importance of the tumour immune microenvironment, including lymphocytes (T cells, B cells, and natural killer (NK) cells), macrophages, dendritic cells (DCs), and neutrophils, is increasingly recognised, as these cells play a critical role in regulating tumour progression and drug response [12,13,14]. Notably, CD8^+^ tumour-infiltrating lymphocytes (TILs) are crucial effectors of antitumour activity, producing interleukin (IL)2, IL-12 and interferon (IFN)-γ, which facilitate the destruction of tumour cells [15,16,17]. High levels of CD8^+^ T cells in the TME are associated with better prognoses and served as a positive predictive marker of survival in CRC patients [18,19].

However, CRC cells can overexpress proteins that facilitate immune evasion, such as the programmed death ligand 1 (PD-L1), which binds programmed death receptor 1 (PD-1) on activated T-lymphocytes. Upon PD-L1 or PD-L2 engagement, PD-1^+^ CD8^+^ T cells may display an exhausted phenotype characterised by reduced proliferation, cytokine production and cytotoxicity, thereby enabling immune evasion by the tumour [20,21]. Immunotherapies targeting the PD-1/PD-L1 pathway have emerged as effective strategies to overcome immune suppression and induce tumour regression in advanced colorectal cancer [22]. Notably, deficient mismatch repair (dMMR)/microsatellite instability-high (MSI-H) tumours, characterized by a high mutational burden and abundant neoantigens, demonstrate significantly improved responses to PD-1/PD-L1 blockade [23]. In contrast, proficient mismatch repair (pMMR)/microsatellite stable (MSS) tumours—which represent the majority of colorectal cancers—typically show limited sensitivity to monotherapy with immune checkpoint inhibitors, highlighting the need for alternative therapeutic approaches in this subgroup [22,24].

The growing understanding of the TME highlights its complexity and critical role in influencing therapy response, allowing the identification of distinct immune subclasses that shape tumour initiation and therapeutic outcomes [25,26,27,28]. Given the critical influence of the immune microenvironment on tumour progression and response to treatment, there is a pressing need for research models that more accurately reflect these in vivo interactions.

In this sense, organoid-immune co-cultures have been established through different approaches depending on the research objective, utilizing either autologous or allogeneic immune components [29,30,31,32]. Autologous systems most closely recapitulate the native tumour–immune interactions [29] but are often limited by restricted tissue availability, particularly when tumour-infiltrating lymphocytes are required. In contrast, allogeneic systems are more accessible but frequently elicit strong immune responses driven by HLA mismatches [33], which can mask tumour antigen-specific effects.

Air–liquid interface (ALI) culture conditions have enabled the generation of complex epithelial and stromal architectures from human inducible pluripotent stem cells [34] and intestinal stem cells [35], effectively recapitulating key aspects of intestinal development and tissue organization. However, one of the most compelling applications of ALI culture arises when using primary human tumour tissues: under these conditions, it is possible to preserve not only epithelial and stromal compartments, but also the native immune infiltrate. This was first demonstrated by Neal et al., who established a 3D ALI system using patient-derived tumour samples that retained embedded native immune cells across multiple tumour types [36,37]. However, despite its potential, the use of ALI-PDOs in CRC remains under-characterised, and its potential advantages over classical organoid culture methods have yet to be fully explored.

Importantly, emerging evidence indicates that immune checkpoint blockade can paradoxically promote the expansion of tumour-resident regulatory T cells (Tregs), thereby contributing to adaptive immune resistance and limiting therapeutic efficacy. This phenomenon has been well documented in preclinical models and suggested by analyses of peripheral blood in treated patients [38,39,40,41], yet direct assessment within intact human tumour microenvironments remains limited. In colorectal cancer, where PD-1 blockade shows minimal efficacy as monotherapy in pMMR/MSS tumours [42], Treg-mediated immune suppression has been proposed as a potential mechanism underlying therapeutic failure and the improved outcomes observed with combined PD-(L)1 and CTLA-4 inhibition [39].

In this study, we conducted a comparative analysis of two organoid-based models using tumour and peritumour samples (healthy tissue located at least 2 cm from the tumour margin) obtained from CRC patient biopsies. The models examined were: (a) Matrigel-based patient-derived organoids (Matrigel-PDOs), which consist solely of epithelial cells, and (b) ALI-PDOs, which integrate both epithelial cells and native resident immune cells. Rather than introducing a new platform, our aim was to assess how each model preserves tumour complexity and informs drug–immune interactions in CRC. Among the immune dynamics explored, we observed a consistent expansion of regulatory T cells (Tregs) following PD-1 blockade, a finding that aligns with its limited efficacy as monotherapy in pMMR colorectal cancers and its enhanced activity when combined with CTLA-4 inhibitors [43]. Overall, our results suggest that ALI-PDOs more faithfully retain features of the original tumour microenvironment and may serve as a complementary platform to classical PDOs for evaluating immunomodulatory effects of therapy in CRC.

## 2. Materials and Methods

### 2.1. Participant Details

Fresh human colorectal tissues were prospectively collected from patients diagnosed with colorectal cancer (CRC) undergoing elective surgery at La Paz University Hospital (Spain). Biopsies were obtained during resection, only from untreated patients who previously signed a written informed consent under the protocol approved by La Paz University Hospital Ethics Committee (PI-1958), in accordance with Spanish ethical regulations and the Helsinki Declaration. Biopsies were firstly received at the Pathological Anatomy department at La Paz Hospital for examination and only when extra tissue was available, without interfering with medical diagnosis, it was transferred to the IdiPAZ Biobank platform and used for the study. The cohort used for the study included 33 treatment-naive patients, including cecum (*n* = 3), ascending colon (*n* = 12), descending colon (*n* = 3), sigmoid colon (*n* = 12), and rectum (*n* = 3). The cohort comprised individuals with three tumour-node-metastasis (TNM) stages, including stage I (*n* = 7), II (*n* = 11), and III (*n* = 14). The TNM stage was not available for one of the patients. PDOs were randomly distributed in groups of seven for the purpose of performing the different analyses included in the study, except for the comparative analysis between co-cultures and ALI-PDOs (*n* = 5) (Appendix A).

### 2.2. Establishment of Matrigel and ALI-PDOs

Patient-derived organoids (PDOs) were established from both peritumoural (healthy tissue, at least 2 cm away from the edge of the tumour) and tumour fresh tissues of resected colon segments from patients diagnosed with colorectal cancer. Matrigel-based PDOs were generated as described by Sato et al. (2011) [10].

Briefly, fresh tissues were washed with cold PBS containing 1X penicillin/streptomycin (10,000 U/mL, ThermoFisher Scientific, Waltham, MA, USA, Cat#15140-122), 1 µg/mL fungizone (ThermoFisher Scientific, Grand Island, NY, USA, Cat#15240062) and 50 µg/mL gentamicin (ThermoFisher Scientific, Grand Island, NY, USA, Cat#15750060). Tissues were disaggregated by incubation with 8 mM EDTA (Merck Millipore, Billerica, MA, USA, Cat#CAS 6381-92-6) at 4 °C for 30 min. Intestinal crypts were isolated by filtering with 70 µm cell strainers (Corning, Steuben County, NY, USA, Cat#352350), embedded in 50% Matrigel Growth Factor Reduced (GFR) Basement Membrane Matrix (Cultek, Madrid, Spain, Cat#356231), and seeded as 50 µL-drops on treated 24-well plates to form dome-shaped structures. Tumour organoids were cultured in IntestiCult™ OGM Human Basal Medium (StemCell Technologies, Vancouver, BC, Canada, Cat#06010) diluted 1:1 in Advanced Dulbecco’s Modified StemCell Eagle’s Medium/F12 (Thermo Fisher Scientific, Cat#11330032). Peritumour organoids were cultured in IntestiCult™ OGM Human Basal Medium with Organoid Supplement Medium (StemCell Technologies, Vancouver, BC, Canada, Cat#100-0190). Both media were supplemented with Y-27632 (ROCK inhibitor, StemCell Technologies, Vancouver, BC, Canada, Cat#72304) and antibiotics (10 U/mL penicillin, 10 mg/mL streptomycin, 1 µg/mL fungizone (ThermoFisher Scientific, Cat#15240062 and 50 µg/mL gentamicin (ThermoFisher Scientific, Cat#15750060)). Cultures were incubated at 37 °C, in 5% CO_2,_ and the medium was replaced every 3 days.

Air–liquid interface (ALI)-PDOs were generated as described by Neal et al. (2018) [37], with minor modifications. Briefly, tumour and peritumour colorectal tissues (0.05 g) were mechanically dissociated into fragments and seeded in a double collagen gel layered (Cellmatrix type I-A, Nitta Gelatin Inc., Osaka, Japan, Cat#631-00651) within a 30 mm, 0.4 mm inner transwell (Merck Millipore, Billerica, MA, USA, Cat#PICM03050) to form a double dish air–liquid culture system. The transwell was placed into an outer 60 mm cell culture dish (Thermo Fisher Scientific, Waltham, MA, USA, Cat#150462) containing IntestiCult™ OGM Human Basal Medium diluted 1:1 in Advanced Dulbecco’s Modified Eagle’s Medium/F12 (DMEM-F12) for the tumour culture and IntestiCult™ OGM Human Basal Medium with Organoid Supplement Medium for the peritumour. In addition to antibiotics, media was supplemented with recombinant human IL-2 (Peprotech, Cranbury, NJ, USA, Cat#200-02-1MG) at 600 IU/mL as a growth factor for T cells. Cultures were maintained at 37 °C, 5% CO_2_, and medium was changed every week.

### 2.3. Live Organoid Imaging, H&E Staining and Immunofluorescence Analysis

The growth and formation of live organoids were monitored using brightfield microscopy (Leica DMI6000 B) during early (5 days) and late (10 days) growth phases. To monitor PDO growth rates, images of the same cultures were acquired after 5 and 10 days of growth. For haematoxylin-eosin (H&E) and immunofluorescence staining of colon tissues and derived organoids, tissue samples and organoid cultures were fixed in 4% paraformaldehyde (Sigma Aldrich, St. Louis, MO, USA, Cat# 30525-89-4) at 4 °C for 16 h. Fixed tissue samples were paraffin-embedded whereas organoids were embedded in OCT. Slides with 5 micrometre-thick sections were used for staining. Histopathological analysis was performed by H&E staining. Immunofluorescence analyses were carried out using the following reagents: Rhodamine Phalloidin TRITC (Thermo Fisher Scientific, Cat#A12379), anti-human Ki67 (Thermo Fisher Scientific Cat # PA5-19462, RRID AB_10981523), anti-human Mucin-2 (Thermo Fisher Scientific, Cat# MA5-32654, RRID:AB_2809931), anti-human Epcam (Abcam, Cambridge, UK, Cat # ab71916), anti-human Pan-Cytokeratin (eBioscience, San Diego, CA, USA, Cat# 41-9003-82), anti-human CD45 (FITC conjugated; Cell Signalling Technologies, Danvers, MA, USA, Cat# 304038, RRID AB_2562050), anti-human CD3 (PerCP conjugated; Cell Signalling Technologies, Cat# 317337, RRID AB_2563405), and anti-human CD68 (Alexa 647 conjugated; Cell Signalling Technologies, Cat# 333819, RRID: AB_2571962). As a secondary antibody, goat anti-rabbit (Alexa Fluor 647 conjugated; Thermo Fisher Scientific) was used. Samples were mounted in DAPI Fluoromount-G (Thermo Fisher Scientific, Cat#D1306) and images were obtained with a Stellaris 8 confocal microscope (Leica Microsystems, Wetzlar, Germany).

### 2.4. Flow Cytometry Analyses of Immune Cell Populations in Tissue and PDO Cultures

For flow cytometry analysis, readily available peritumour and tumour colon samples, and derived ALI-PDOs (n = 7) were minced and digested with 200 units/mL collagenase IV (Worthington, Cat#NC9836075) and 0.1 mg/mL DNAse I (Sigma-Aldrich, Cat#AMPD1-1KT) at 37 °C for 15 min. Single-cell suspensions were generated by filtering through a 70 µm cell strainer. Cells were stained with LIVE/DEAD™ Fixable Blue (ThermoFisher, Cat#L23105) and Human TruStain FcX™ (ThermoFisher, Cat#L34957) to exclude nonviable cells and block nonspecific antibody binding. A deep profiling of the immune infiltrates was performed using a custom antibody cocktail (Appendix A). Data was acquired on an Aurora full-spectrum flow cytometer (Cytek^®^ Biosciences Inc., Fremont, CA, USA) and analysed using FlowJo v10.6.2 software. The characterization of the immune infiltrate was carried out following a gating strategy validated in the laboratory (Appendix A).

### 2.5. Soluble Immune Checkpoint (IC) and Cytokine Detection Assay in PDO Supernatants

Seven-day cell culture supernatants from matched tumour Matrigel and ALI-PDOs derived from 7 patients were collected in order to analyse the presence of soluble immune checkpoints (ICs) and their ligands (ICLs). For ICs quantification supernatants were processed according to the protocol provided with The LEGENDplex™ HU Immune Checkpoint Panel 1 (Biolegend, San Diego, CA, USA, Cat#740869) that allows for simultaneous quantification of 12 key immune checkpoint proteins: sCD25 (IL2-Ra), 4-1BB, sCD27, B7.2 (CD86), free active TGF-β1, CTLA-4, PDL-1, PDL-2, PD-1, Tim-3, LAG-3, and Galectin-9. For cytokine quantification supernatants were processed according to the protocol provided with the The LEGENDplex™ Human Essential Immune Response Panel (13-plex) (Biolegend, Cat# 740930) that allows for simultaneous quantification of 13 human cytokines, including IL-4, IL-2, CXCL10 (IP10), IL-1β, TNF-α (TNFSF2), CCL2 (MCP-1), IL-17A, IL-6, IL-10, IFN-γ, IL-12p70, TGF-β1 (Free Active), CXCL8 (IL-8). Prepared samples were acquired on an Aurora full-spectrum 5L flow cytometer (Cytek^®^ Biosciences Inc., Fremont, CA, USA) and analysed using FlowJo v10.6.2 software (Ashland, OR, USA). To assess relative immunomodulatory programs, we performed log2-transformation of each parameter followed by within-sample Z-score normalization across analytes.

### 2.6. Whole Exome Sequencing (WES) and Genomic Analysis

Genomic DNA was extracted from snap-frozen patient tissue and 14-day culture-derived Matrigel and matched ALI-PDOs (n = 7), using a NZY Tissue gDNA Isolation kit (NZYtech, Lisbon, Portugal). Whole Exome Sequencing (WES) was performed using the Sureselect V8 + NovaSeqX 150 PE (150 × 2 bp) 6 Gb/sample by Macrogen (San Diego, CA, USA). Seven tumour Matrigel and ALI-PDOs were sequenced with matched tumour tissues for mutation concordance analysis.

Unless otherwise stated, the human genome version hg38 with alternative contigs removed was used for all analyses and annotations were taken from Ensembl version 111 [44]. All programs from the GATK suite [45] used correspond to version 4.5.0.0. Somatic variants were obtained according to GATK best practices [46,47]. For computational execution of tasks in parallel, CGAT-core was used [48].

Briefly, in a read cleaning and filtering stage, raw paired-end Whole Exome Sequencing reads were quality controlled (Appendix A) using FastQC version 0.12.0 [49]. DNA-seq adapters were removed using Cutadapt version 4.7 [50] in paired-end mode. Only read pairs with both single-end fragments remaining were kept. The processed paired-end DNA-seq reads were mapped using BWA version 0.7.17 [51] in paired-end mode to the human genome using the BWA-MEM algorithm, marking shorter split hits as secondary and establishing sample read groups, the remaining options were left on default.

Duplicates were marked using Picard Markduplicates from the GATK suite, the NM, MD, and UQ tags from the resulting alignment files were fixed using SetNmMdAndUqTags (GATK). Using these mappings, base quality recalibration was performed first using BaseRecalibrator function from GATK to build the model with the known-sites option referencing the database of known sites from the Broad Institute (ftp.broadinstitute.org/bundle/hg38/dbsnp_146.hg38.vcf.gz, accessed on 24 July 2025). The model is then applied to adjust the base quality scores using the ApplyBQSR function from GATK on the mappings file.

Variant calling was performed to obtain variants present in organoids and not in the raw tumour by using Mutect2 (GATK) with raw tumour as pseudonormal samples and organoids as tumour with a public panel of normals generated from the 1000 Genomes Project [52]) to account for sequencing artifacts and germline allele frequencies from germline variants from the GATK best practices repository. LearnReadOrientationModel, GetPileupSummaries and CalculateContamination and FilterMutectCalls (GATK) were used according to best practices to add filtering tags and a verification and curation step was performed using a personalized Python (v3.8.13) script with the help of the Cyvcf2 library version v0.31.1. [53].

The process was repeated, switching pseudonormal and tumour samples to obtain raw tumour exclusive variants. For each comparison, variants found to be present in both conditions were merged and removed from the state-exclusive variants using BCFtools merge and isec, respectively. In a final step, variants containing variant allele frequency below 0.05 or read depth below 10 reads, were disregarded according to best practices [54,55,56].

### 2.7. Co-Culture of Tumour Organoids with Autologous PBMCs

Ten millilitres of peripheral blood were collected at the same time that tumour specimens were collected. PBMCs were isolated using Ficoll-Paque Plus (Merck, Darmstadt, Germany, Cat#GE17-1440-03) density gradient separation and were cryopreserved until later use.

One day before co-culture, PBMC were thawed in pre-warmed (37 °C) and seeded on 6-well plates at 2 × 10^6^ cell/well with RPMI 1640 medium (ThermoFisher Scientific Cat# 11875093) supplemented with 1X penicillin/streptomycin (10,000 U/mL, ThermoFisher Scientific Cat#15140-122), 10% FBS (Fetal bovine serum, ThermoFisher Scientific, Cat#A5670701), recombinant human IL-2 (Peprotech Cat# 200-02-1MG) at 600 IU/mL, and incubated at 37 °C overnight.

On the day of co-culture, intact tumour organoids were isolated from Matrigel using cold PBS. Part of the organoids were dissociated from single cells with TrypLE Express (ThermoFisher Scientific Cat# 12604013) and counted using a hemocytometer. This was used to infer the number of tumour cells per tumour organoid to allow co-culture of organoids and autologous PBMCs cells at a 5:1 (PBMC:organoid cell) ratio, as previously established in the literature [57,58]. The cell mixture was embedded in 30% Matrigel Growth Factor Reduced (GFR) Basement Membrane Matrix (Cultek, Madrid, Spain, Cat#356231), and seeded as 10 µL-drops on treated 96-well plates to form dome-shaped structures.

Co-cultures were covered with a 1:1 mix of IntestiCult™ OGM Human Basal Medium [(StemCell Technologies, Vancouver, BC, Canada, Cat#06010) (diluted 1:1 in Advanced Dulbecco’s Modified StemCell Eagle’s Medium/F12 (Thermo Fisher Scientific, Waltham, MA, USA, Cat#11330032)] and RPMI medium. The co-culture media was supplemented with antibiotics, Y-27632 ROCK inhibitor, and 600 U/mL IL-2. For each experimental condition, three technical replicates were performed, and the samples were subsequently pooled for flow cytometry analysis to obtain a representative measure while minimizing intra-assay variability.

### 2.8. 5-Fluorouracil and Anti-PD-1 Assays

To perform viability assays and immune cell modulation studies in tumour Matrigel-based PDOs, confluent cultures were harvested with DMEM-F12 medium by centrifugation at 440× *g* for 3 min and seeded on a 96-well plate at 100 organoid/well and incubated for 24 h in a 5% CO_2_ incubator at 37 °C before treatment. Matrigel-based PDOs were exposed to 100 µM concentrations of 5-FU. The cytotoxic assays in tumour ALI-based PDOs were performed by adding the same concentrations of 5-FU as for Matrigel-PDOs after 3 days of culture, when organoid structures begin to form. PDOs were culture in the presence of the drug for 5 days. ALI-PDOs and Matrigel co-cultures used for anti–PD-1 assays were supplemented with organoid medium containing 10 µg/mL nivolumab (BristolMyers Squibb, Princeton, NJ, USA, Cat#A2002). IgG controls (nontreated) were included for comparison. Medium was replaced every 3 days.

After 5 days in the presence of the drug, PDOs were photographed and collected to perform cytotoxicity assays and a profiling of the immune infiltrates. Matrigel-PDOs were digested with TrypLE Express (Gibco, Grand Island, NY, USA, Cat#12605010) and ALI-PDOs with 200 units/mL collagenase IV (Worthington, Columbus, OH, USA, Cat#NC9836075) and 0.1 mg/mL DNAse I (Sigma-Aldrich, St. Louis, MO, USA, Cat#AMPD1-1KT) and stained with LIVE/DEAD™ Fixable Blue and Human TruStain FcX™ (ThermoFisher, Waltham, MA, USA, Cat#L34957). PerCP anti-human CD45 (Biolegend, San Diego, CA, USA, Cat# 368506 was also added to assess viability of immune infiltrate populations. In addition, a deep profiling of the immune infiltrates was performed using a custom antibody cocktail (Appendix A).

### 2.9. Quantification and Statistical Analysis

Statistical analysis was performed on Prism (GraphPad Software v10, Boston, MA, USA). When indicated, statistical significance was evaluated by paired Student’s *t*-tests. Significance values are indicated as * *p* < 0.05, ** *p* < 0.01, *** *p* < 0.001, **** *p* < 0.0001.

In the heatmaps for PDO immune profiles, we used the following immune descriptors for each sample: percentage of CD45 within total; the percentage of αβ T cells, CD4 T cells, CD8 T cells, CD4^+^CD8^+^ T cells, CD4^−^CD8^−^ T cells, CD4 Treg, NKT-like, γδ T cells, B cells, cDC1, cDC2, NK cells, ILCs, neutrophils, monocytes and TAMs within CD45. We evaluated their similarity using the Pearson correlation coefficient.

## 3. Results

### 3.1. Comparative Structural and Cell Type Characterization of Matrigel and ALI-PDOs

Readily available biopsies obtained during surgery from 33 patients diagnosed with colorectal cancer (CRC) were obtained from two areas: tumour and peritumour, the latter being healthy tissue at least 2 cm away from the edge of the tumour. The clinicopathological characteristics of CRC patients are shown in Appendix A. Organoids were generated from each patient following two different protocols: Matrigel-based patient-derived organoid (Matrigel-PDO) and air–liquid interface patient-derived organoid (ALI-PDO). Subsequently, a thorough and comparative characterisation was conducted to assess the potential advantages of each system in relation to CRC disease modelling.

Figure 1 illustrates the structural and cellular composition of Matrigel- and ALI-PDOs generated from biopsies obtained from patient #5 (Appendix A). As Figure 1A shows, organoids were generated from colorectal cancer patient biopsies. Representative haematoxylin and eosin (H&E) staining of the original tumour tissue and matched peritumour tissue are shown in Figure 1B,C, respectively.

Matrigel-PDOs generated from intestinal crypts (Figure 1D) of both tumour and peritumour tissues were analysed. In tumour PDOs, a range of patient-specific morphologies were observed, from thin-walled cystic structures to compact spherical structures (Figure 1E, upper panels). In the case of peritumour Matrigel-PDOs, complex multi-lobular structures presenting crypt-villus domains were observed (Figure 1E, lower panels). In addition, H&E staining of Matrigel-PDOs showed that they retained the histopathologic morphology of their tissue of origin. Tumour Matrigel-PDOs had large, deep-stained nuclei, irregular arrangement, and decreased cytoplasm (Figure 1F, upper panel). Peritumoural Matrigel-PDOs preserved cytological features, including the architecture of colon epithelium with villi and crypt-like domains (Figure 1F, bottom panel).

In the case of ALI-PDOs, mechanically dissociated tissue fragments were embedded in type I collagen and cultured in a transwell-based air–liquid interface system (Figure 1G). After 10 days of culture, small buds progressively expanded, and the number of organoid colonies increased accordingly. Importantly, tumour ALI-PDOs were generally larger than peritumour ALI-PDOs (Figure 1H). Additionally, H&E staining of ALI-PDOs showed their ability to retain the histopathologic features of their parental tissues better than Matrigel-PDOs. ALI-PDOs consist of a monolayer of epithelial-like cells and the surrounding stromal-like cells that conform the tumour microenvironment-like structure (Figure 1I), not observed in Matrigel-PDOs.

To verify that organoids contained progenitor cells and represent human disease, the expression of classic colon markers was analysed in Matrigel and ALI-PDOs along with their respective primary tissues. Both, Matrigel-PDOs and ALI-PDOs from tumour (Figure 1J) and peritumour (Figure 1K) showed Ki67 labelling, a proliferation marker highly expressed in cycling cells [59]. Additionally, the expression of Mucin-2, a specific goblet cell marker, was assessed. Elevated Mucin-2 levels were observed in peritumour colon organoids in both Matrigel-PDOs and ALI-PDOs (Figure 1K), while most tumour organoids exhibited negative or mildly positive staining for this marker (Figure 1J). However, organoids derived from mucinous adenocarcinoma (patient #3, Appendix A) retained Mucin-2 expression, reflecting preservation of the mucinous phenotype in vitro. This finding aligns with previous reports [60,61,62] indicating that patient-derived organoids (PDOs) can faithfully recapitulate key histopathological and molecular features of the primary tumour, including goblet cell differentiation and mucin secretion.

The epithelial cell identity of the generated PDOs was also confirmed by assessing the consistent expression of Pan-Cytokeratin (PanCK) and EpCAM across other patients of the cohort (Appendix A, respectively).

Our data indicate that both methods effectively generate organoids that mimic the original tissue’s structure and disease state (morphological and pathological characteristics).

### 3.2. ALI-PDOs Reproduce the Complexity of the Immune Microenvironment of Original Tissues

While ALI-PDOs successfully mimic tissue architecture and pathology, their potential to accurately model the TME has yet to be thoroughly investigated in CRC [37,63,64]. The identification and characterization of immune cell populations within ALI-PDOs are essential for deepening our understanding of tumour–immune interactions and for assessing responses to chemotherapy and immunotherapy.

Thus, we examined the TME within tumour-derived ALI-PDOs using a multifaceted approach. This involved immunofluorescence staining with specific cell surface markers to visualise immune cell populations, full-spectrum flow cytometry to identify and quantify immune infiltrates within the organoids, and a comparative analysis with their corresponding primary tissues. Additionally, we quantified soluble factors involved in the immune response, providing a comprehensive understanding of the immune landscape in these organoids.

Flow cytometry analysis revealed that, unlike Matrigel-PDOs, ALI-PDOs successfully retained their native immune cell infiltrates. After 7 days of culture, ALI-PDOs exhibited a significant CD45^+^ cell population, a feature notably absent in Matrigel-PDOs (Figure 2A). Furthermore, immunofluorescence staining confirmed the retention of native immune infiltrates within ALI-PDOs, as evidenced by positive labelling for CD45, CD3, and CD68 at this time point (Figure 2B–D).

Moreover, significant differences were observed in the relative release of soluble immune mediators between ALI- and Matrigel-PDOs, as assessed by log-transformed values followed by within-sample Z-score normalization across analytes. Using this approach, ALI-PDOs exhibited a selective relative enrichment of IL-6 and MCP-1 compared with Matrigel-PDOs (Figure 2E,F), whereas several other mediators, including Galectin-9, IFN-γ, and IL-12p70, displayed relative decreases (Figure 2G–I). Soluble markers showing no significant relative differences between culture systems are shown in Appendix A. Collectively, these relative profiles indicate a selective enrichment of inflammatory and chemokine-associated signals in ALI-PDOs.

To further delineate the immune cell composition within ALI-PDOs and compare it to the original TME, we utilised a full-spectral cytometry panel (Appendix A). This comprehensive analysis enabled the evaluation of matched pairs of ALI-PDOs and their corresponding tumour and peritumour tissues from CRC patients included in the study.

As illustrated in Figure 3A,B, the percentage of CD45^+^ cells in live cells across both tumour and peritumour tissues, as well as their respective ALI organoids, ranged from 0.5% to 56.8%, showing substantial inter-tumoural heterogeneity, with no significant differences observed between the original tissues and matched organoids (Figure 3B). In line with the observed infiltration of immune cells, after 7 days of culture, ALI-PDOs retained a subset of immune cell types from the original tissue (Figure 3C,D). This included lymphoid lineages, such as αβ T cells, γδ T cells, B lymphocytes, and innate lymphoid cells (ILCs) (Figure 3E–H).

There was also preservation of myeloid populations, such as conventional type 1 dendritic cells (cDC1), conventional type 2 dendritic cells (cDC2), NK cells, neutrophils, monocytes, and macrophages (Figure 3I–N). We represented the similarity of the immune composition across fresh tissue samples and their matched ALI-PDOs for both, tumour and peritumour samples (Figure 3O,P). We observed some heterogeneity when comparing peritumour samples and their derived ALI organoids; however, the concordance between primary tissues with their respective corresponding organoids was largely preserved across patients (Figure 3P). On the other hand, tumour ALI-PDOs exhibited a higher degree of immune profile similarity to their primary tumours compared to peritumour ALI-PDOs (Figure 3O).

Note that, despite maintaining the general makeup of the immune system, a shift in the proportions of αβ T cells and B cells, was observed in tumour and peritumour ALI-PDOs, respectively (Figure 3F,G). In the latter, a lower frequency of B cells was detected in peritumour ALI-PDOs compared to the original tissue. In the case of the discrepancy observed in αβ T cells, tumour ALI-PDOs displayed a noteworthy enrichment of total αβ T cells compared to the original tumour tissue (Figure 3F). To understand which specific T cell subsets were driving this increase, we further characterised them based on the expression of CD4 and CD8, classical markers that distinguish helper and cytotoxic T cells, respectively.

Our analysis revealed a distinct distribution of T cell subsets within ALI-PDOs. CD4^+^ T cells were the primary contributors to the overall rise in αβ T cells, showing a significant increase within ALI-PDOs (Figure 4A). This increase was not attributable to the regulatory CD4 T cell subset, whose levels were preserved in ALI-PDOs (Figure 4B). Notably, levels of CD8^+^ T cells remained unchanged compared to their original tissue (Figure 4C). Interestingly, double-positive T cells, expressing both CD4 and CD8, mirrored the CD4^+^ T cell trend, with a statistically significant rise in ALI-PDOs (Figure 4D), not observed in double-negative T cells (Figure 4E).

A crucial aspect of in vitro/ex vivo immune checkpoint blockade models is replicating the exhaustion state of tumour-infiltrating CD8^+^ T cells. To assess this, we investigated the co-expression of exhaustion markers PD-1 and CD38 on CD8^+^ T cell subsets (Appendix A) from tumour tissues and matched ALI-PDOs. Importantly, ALI-PDOs not only preserved this co-expression pattern but also displayed similar variations across different T cell differentiation stages, classified into naïve, CD45RA^+^ terminal effector, memory, early effector, early-like effector, intermediate effector, and terminal effector, effectively mirroring the exhaustion profile of the original tumour (Figure 4F,G). This remarkable ability to recapitulate exhaustion highlights the potential of ALI-PDOs as a powerful platform for patient-specific evaluation of immune checkpoint inhibitor efficacy.

In addition, we characterised the membrane expression of selected immune checkpoints corresponding to soluble factors detected in supernatants, including PD-1, across different cell types detected by flow cytometry. Notably, CD25, CD27, and PD-1 were expressed at significantly higher levels in immune cells—particularly within lymphocyte populations—compared with CD45^−^ cells (Appendix A). In contrast, PDL-1 expression was predominantly associated with myeloid cells, although a substantial proportion of CD45^−^ cells also displayed PDL-1 positivity (Appendix A). These findings were consistent with the Human Colon Cancer Atlas (c295) single-cell transcriptomic dataset, which revealed similar patterns of checkpoint molecule distribution across cell subsets (Appendix A). Together, these results support the capacity of ALI-PDOs to preserve physiologically relevant patterns of immune checkpoint expression.

### 3.3. PDOs Faithfully Preserve the Mutational Landscape of Their Original Primary Tumours

To determine whether colorectal cancer Matrigel and ALI-PDOs cultured for 14 days retained the genetics of the original tumours, whole-exome sequencing (WES) was performed on matched tumour tissue, Matrigel-PDOs and ALI-PDOs from 7 patients (#8 to #14). Sequencing depth and genome coverage were comparable across all conditions, and variant calling was consistent with established colorectal cancer mutational loads (Appendix A).

When directly comparing the two culture systems, Matrigel-PDOs showed significantly greater deviation from the original tissue mutational profile compared to ALI-PDOs (Figure 5A). In six out of seven cases, ALI-PDOs more closely matched the tissue mutational load. Across the cohort, Matrigel-PDOs showed higher variants-per-coverage values (Appendix A) and most additional variants detected in PDOs but not in the tissue were specific to the Matrigel condition (Figure 5B).

These findings suggest that while Matrigel-PDOs can effectively recapitulate tumour genetics as established in the literature [65], they may be more prone to culture-related variability in mutational detection, potentially due to enhanced tumour cell enrichment and clonal expansion effects during extended culture periods.

Across patients, both PDO formats preserved the main somatic driver alterations identified in the original tumours, including mutations in *APC*, *TP53*, *KRAS*, *PIK3CA*, *FBXW7*, and *AKT1*. The frequency of these alterations matched expectations from large-scale CRC sequencing studies [66,67,68]. As shown in Figure 5C–E, these driver mutations were retained in the majority of cases in both culture systems, but the alteration rates per biopsy sample varied widely, with SNP missense mutations being the most common (46.2%). This observation aligns with previous studies, which have demonstrated that missense mutations are the most prevalent type of CRC and are the primary drivers of tumour development [69,70,71].

Reverting mutations to wildtype in ALI-PDOs (n = 976) and Matrigel-PDOs (n = 961) exhibited highly similar variant allele frequency (VAF) distributions, with no statistically significant difference (Mann–Whitney U test, *p* = 0.85) (Appendix A). Exclusion of low-frequency variants (<5% VAF) did not materially affect reversion rates or overall conclusions, indicating that reversions are not attributable to rare tumour subclones.

Overall, this analysis supports that both Matrigel- and ALI-PDOs adequately preserve the key genetic features of their parental colorectal tumours, consistent with the expected fidelity of PDO platforms. Given the limited cohort size, these genomic results are presented as quality-control evidence for genetic drift.

### 3.4. ALI-PDOs Provide a Platform to Explore the Antitumour and Immunomodulatory Effects of Cancer Therapies

We next explored the potential of ALI-PDOs to model the interplay between cancer therapies and the tumour microenvironment. Matrigel and ALI-PDOs derived from colorectal tumour samples were exposed to 100 µM of 5-fluorouracil (5-FU), a standard-of-care chemotherapeutic agent for CRC patients [72]. After 5 days of exposure to the drug, the response to 5-FU varied remarkably in Matrigel-PDOs, with those from patients #16, #20 and #21 displaying low sensitivity to the 5-FU (represented as CD45^−^) (Figure 6A). Some PDOs, such as #18, showed a dramatic loss of their structure (Figure 6B, central panel) compared to control (nontreated) organoids (Figure 6B, left panel). However, some PDOs, such as #21, displayed a nonresponsive pattern confirmed by bright-field microscopy images (Figure 6B, right panel).

Similarly, the responsiveness of the CD45^−^ compartment from ALI-PDOs to 5-FU exhibited a wide variability across samples (Figure 6C,E). ALI-PDOs from patients #15, #18, #19, #20 and #21 exhibited a 5-FU sensitivity pattern similar to that of their Matrigel-PDO counterparts. In contrast, CD45^−^ cells from patient #16 ALI-PDO became sensitive in the presence of an intact TME, while those from patient #17 turned unresponsive (Figure 6C). In contrast, CD45^+^ immune cells from ALI-PDOs demonstrated a higher degree of sensitivity to 5-FU (Figure 6D).

Alternatively, we also explored whether ALI-PDOs could model the immune dynamics triggered by PD-1 blockade with nivolumab, one of the most widely utilized immune checkpoint inhibitors [73,74]. On the one hand, we found that nivolumab mildly decreased the number of CD45^−^ tumour cells (Figure 6F), possibly owing to a clinically irrelevant effect as monotherapy in pMMR/MSS CRC [42]. On the other hand, nivolumab clearly promoted an enrichment in CD45^+^ immune infiltrates in colorectal cancer ALI-PDOs (Figure 6G), which was especially driven by CD3^+^T cells (Figure 6H).

We sought to further elucidate whether ALI-PDOs could reveal the differential effects of 5-FU and nivolumab on the immune cell landscape within the ALI-PDO model. Treatment with 5-FU led to a marked depletion of CD45^+^ cells, primarily driven by a significant reduction in αβ T cells, particularly CD8 T cells (Figure 6I). In contrast, nivolumab stimulated the expansion of CD3^+^ T cells, predominantly by promoting the proliferation of conventional αβ T cells, specifically CD4 T cells, among which CD4 Treg cells were particularly prominent (Figure 6J).

To extend our analysis, we recruited five additional CRC patients to directly compare the immune composition of ALI-PDOs with the widely used Matrigel-based autologous co-culture system with PBMCs. PCA and cosine similarity analyses revealed that ALI-PDOs more faithfully preserved the immune infiltrates present in the original tumours than Matrigel co-cultures (Figure 7A,B). We next assessed the ability of both models to reproduce immune dynamics in response to in vitro treatments. Consistent with our previous observations (Figure 6J), exposure to nivolumab promoted a consistent expansion of Treg cells within ALI-PDO infiltrates (Figure 7C). In contrast, Matrigel co-cultures failed to recapitulate this phenomenon (Figure 7D), suggesting that autologous PBMC-based systems may not preserve the contextual signals required to support conserved immunomodulatory responses.

These findings highlight CRC ALI-PDOs as a powerful platform for investigating drug-induced immune dynamics, offering valuable insights into potential combination therapies.

## 4. Discussion

Patient-derived organoids (PDOs) have gained increasing recognition as powerful tools in cancer research and therapeutic development [65,75,76,77,78,79]. This study aimed to determine whether air–liquid interface organoids (ALI-PDOs), derived from colorectal cancer (CRC) patient samples, offer advantages over conventional Matrigel-based PDOs (Matrigel-PDOs), not only in faithfully replicating histopathological features but, more importantly, in preserving the tumour microenvironment (TME) and the genetic characteristics.

As anticipated, Matrigel-PDOs consist solely of neoplastic epithelium and the use of peripheral blood mononuclear cells (PBMCs) for co-culture presents inherent limitations, as circulating immune cells differ substantially from tumour-resident populations in their activation and exhaustion states and the expression of actionable molecular targets. Moreover, achieving physiologically relevant immune-to-tumour cell ratios remains challenging [29,30]. Consequently, PBMC–Matrigel-PDO co-cultures often contain nonexhausted tumour-reactive T cells capable of controlling organoid growth [80]. Co-culture systems are highly valuable for investigating intercellular interactions—often employing T cells pre-activated with IFN-γ [81] or CD28 antibodies [57]—and are convenient for cytotoxicity assays [80,82]. However, they fail to reproduce the immune-to-tumour cell density observed in vivo—which varies according to tumour type, anatomical location, and disease stage—and the metabolic and functional fitness of tumour-native immune cells [83].

In contrast, ALI-PDOs provide a promising alternative by maintaining the endogenous immune compartment of tumours in vitro, thereby preserving the immune exhaustion state characteristic of primary tumours. Although Neal et al. demonstrated the utility of the ALI system across multiple tumour types—including lung, renal, melanoma, pancreatic, and thyroid cancers—their assessment of colorectal cancer samples was limited, leaving substantial scope for further investigation [37].

Despite numerous studies that have examined the ability of ALI and Matrigel-embedded organoid cultures to recapitulate their tissue of origin [9,10,34], most have primarily focused on comparisons between two-dimensional monolayer cultures [11,64]. While these investigations are undoubtedly relevant, our study advances this work by directly comparing three-dimensional Matrigel- and ALI-PDO models using patient-matched samples.

We confirmed that both Matrigel- and ALI-based systems effectively retained the histopathological features of their parental tissues. However, ALI-PDOs demonstrated a superior preservation of architectural complexity and phenotypic diversity of the original colon tissue, encompassing both tumour and adjacent nontumour regions. Notably, ALI-PDOs exhibited greater structural intricacy, where stroma-like cells were observed within the crypt-villus domains—features absent in Matrigel-PDOs—thereby offering a more physiologically relevant model of the TME.

The preservation of both epithelial and immune compartments within ALI-PDOs provides a unique opportunity to examine the immunomodulatory landscape of human CRC in a controlled ex vivo setting. To characterise this landscape, we analysed soluble immune checkpoints, checkpoint ligands, and cytokines released into the culture supernatants. Rather than indicating a uniform increase in immunomodulatory factors, relative profiling of soluble mediators revealed a selective enrichment of specific inflammatory and chemokine-associated signals in ALI-PDOs. In particular, IL-6 and MCP-1 emerged as the predominant relatively enriched mediators, whereas several immune checkpoint–associated or lymphoid-related factors, including Galectin-9, IFN-γ, and IL-12p70, showed relative decreases. This pattern suggests that the soluble immune milieu generated by ALI-PDOs reflects an inflammatory and chemotactic program rather than a global activation of immune checkpoint pathways, consistent with the presence of innate and myeloid components within the preserved tumour microenvironment.

Flow cytometry further confirmed the presence of diverse lymphoid and myeloid populations within ALI-PDOs. We observed an overrepresentation of conventional αβ T cells within tumour-derived ALI-PDOs compared to their matched fresh tissues. This expansion is likely driven by supplementary IL-2, a critical T cell growth factor. Notably, the overall proliferation of conventional αβ T cells was primarily dependent on CD4^+^ T cells. However, this expansion could not be attributed to Treg cells, despite their high expression of the IL-2 receptor CD25. Tumour-infiltrating CD4^+^ T cells, irrespective of FOXP3 expression, exhibited elevated CD25 levels, suggesting an activated state [84]. Although IL-2 is typically associated with CD8^+^ T cell proliferation during initial TCR activation [85], it also plays a crucial role in the long-term survival of activated CD4^+^ T cells during the contraction phase of the immune response [86]. Further research is required to determine the influence of additional factors within the organoid culture medium on the exhaustion state of specific T cell subsets.

Despite the inherent limitations of ex vivo models, ALI-PDOs successfully recapitulate key aspects of the complex TME. We identified diverse myeloid populations, including neutrophils, monocytes, macrophages, and conventional dendritic cells (cDCs), all of which play essential roles in regulating CD8^+^ T cell activation and responsiveness to immune checkpoint blockade therapies [87,88,89,90]. Additionally, the CD8^+^ T cells infiltrating ALI-PDOs retained PD-1 and CD38 expression profiles consistent with their primary tissues [91,92], reinforcing the potential of ALI-PDOs as robust models for studying immune responses to anti–PD-1 therapies.

Heatmap correlation analysis revealed that tumour-derived PDOs more accurately preserved the immune profile of their corresponding fresh tissues compared to peritumour PDOs, a finding of significant relevance for CRC disease modelling. Furthermore, ALI-PDOs more faithfully preserve the immune infiltrates of the original tumours compared with the widely used Matrigel-based autologous co-culture system with PBMCs, underscoring their advantage as a model for assessing immunotherapy responses.

To assess whether colon PDOs accurately reflected the genomic landscape of their corresponding primary tumours, we performed whole-exome sequencing (WES). Both ALI- and Matrigel-PDOs retained mutations in key CRC-associated genes, including *APC* and *TP53*—the two most frequently altered genes in CRC [68]—which drive dysregulation of the WNT pathway and contribute to tumour progression [68,93,94]. Additional mutations in *KRAS*, *PIK3CA*, *FBXW7*, and *AKT1* were also consistently identified, with mutation frequencies and copy number alterations aligning well with those reported in large clinical cohorts [68]. However, while ALI-PDOs closely mirrored the mutational burden of the original tissues, Matrigel-PDOs showed a tendency to acquire additional mutations, including within key oncogenes and tumour suppressors. These differences may result from the in vitro expansion of subclonal populations or the accumulation of culture-induced alterations, which were more pronounced in Matrigel-PDOs. Although most of these mutations were silent, they could still affect gene expression or reflect increased genomic instability. In contrast, the reduced mutation acquisition observed in ALI-PDOs may be attributed to selective immune pressure on tumour neoantigens, consistent with the concept of immunoediting, where immune surveillance influences tumour evolution [95,96]. The process of immunoediting, in which tumour stroma exerts immune pressure on genetic variants, may contribute to the superior genetic fidelity of ALI-PDOs, further reinforcing their utility in biomedical research.

Despite the advantages of ALI-PDOs in preserving endogenous tumour–immune interactions, several inherent limitations of this platform should be acknowledged. Unlike Matrigel-based PDOs, ALI cultures are not designed for long-term expansion, passaging, or cryopreservation, as immune infiltrates are progressively lost over time, which restricts experimental scalability and repeatability [37]. Their dependence on freshly resected surgical tissue further limits throughput and broad applicability. In addition, because ALI-PDOs maintain heterogeneous epithelial, stromal, and immune compartments, comprehensive molecular characterization at the initiation of culture is not always feasible, particularly when tissue availability is limited. These constraints underscore that ALI-PDOs should not be viewed as a replacement for conventional Matrigel-based organoids, but rather as a complementary system optimally suited for short-term functional interrogation of native tumour–immune ecosystems, whereas Matrigel-PDOs remain advantageous for scalable, genetically defined, and highly reproducible studies.

Given the critical role of chemotherapy in CRC treatment [97], particularly with 5-FU, we investigated its effects on both Matrigel- and ALI-based tumour organoids. Both Matrigel- and ALI-PDOs displayed a wide range of responses to 5-FU. Our data suggest that the immune heterogeneity retained in ALI-PDOs—including macrophage subsets, immune cell-derived exosomes and soluble immunomodulatory agents, such as cytokines, chemokines and immune checkpoint ligands—may be important for reproducing patient-specific sensitivity to chemotherapy [98]. Additionally, 5-FU diminished the presence of CD4^+^ T cells and CD8^+^ T cells within ALI-PDO microenvironment. This observation aligns with previous reports of a transient lymphocyte depletion following neoadjuvant therapy [99,100].

Expanding the benefits of immunotherapy to CRC—particularly pMMR/MSS tumours—remains a clinical challenge. In our study, PD-1 blockade with nivolumab reshaped the immune microenvironment in ALI-PDOs, though changes were largely patient-specific. General T cell dynamics were not consistent across patients, a variability also noted in tumour-derived organoid platforms by Neal and colleagues [37]. A consistent finding, however, was the enrichment of pre-existing Tregs following PD-1 blockade. This phenomenon aligns with recent reports in mouse models [38,101], whereas human data remains largely restricted to peripheral blood analyses, with little direct tumour-tissue validation [39,40]. Mechanistically, Treg expansion may be driven by increased IL-2—from activated CD8^+^ T cells or others—ICOS upregulation on Tregs, and possibly release from suppression via checkpoint blockade [39,41]. Importantly, Matrigel co-cultures with PBMCs did not reproduce this enrichment, maybe due the lack of key stromal components involved in Treg proliferation on survival or, the distinct metabolic state, antigen specificity, and functional fitness of tumour-resident versus circulating lymphocytes [102]. Clinically, PD-1 blockade only showed a biologically irrelevant control of the epithelial/tumour compartment, which is consistent with its objective response rates (ORRs) of only 0–5% as monotherapy in pMMR CRC [42]. The preferential expansion of Tregs under PD-1 blockade may represent a compensatory mechanism of immune suppression, helping to explain the limited efficacy of monotherapy. Conversely, this mechanism may underlie the improved outcomes seen with combination immunotherapy, as dual PD-(L)1 and CTLA-4 blockade can achieve ORRs of up to 20% in pMMR CRC patients, supporting the rationale for targeting multiple immune-regulatory axes [43].

## 5. Conclusions

This study demonstrates that ALI-PDOs faithfully reproduce the structural, genomic, and immune features of primary CRC, providing a robust platform to explore patient-specific tumour–immune dynamics. Notably, the response patterns we observed with immune-checkpoint blockade resemble the limited efficacy reported clinically for anti–PD-1 monotherapy—particularly nivolumab—which may relate to the expansion of regulatory T cells seen ex vivo and could help explain why nivolumab combined with ipilimumab achieves superior outcomes. Together, these findings support the utility of ALI-PDOs for dissecting drug–immune interactions and informing more precise therapeutic strategies for colorectal cancer.

## Figures and Tables

**Figure 1 cancers-18-00132-f001:**
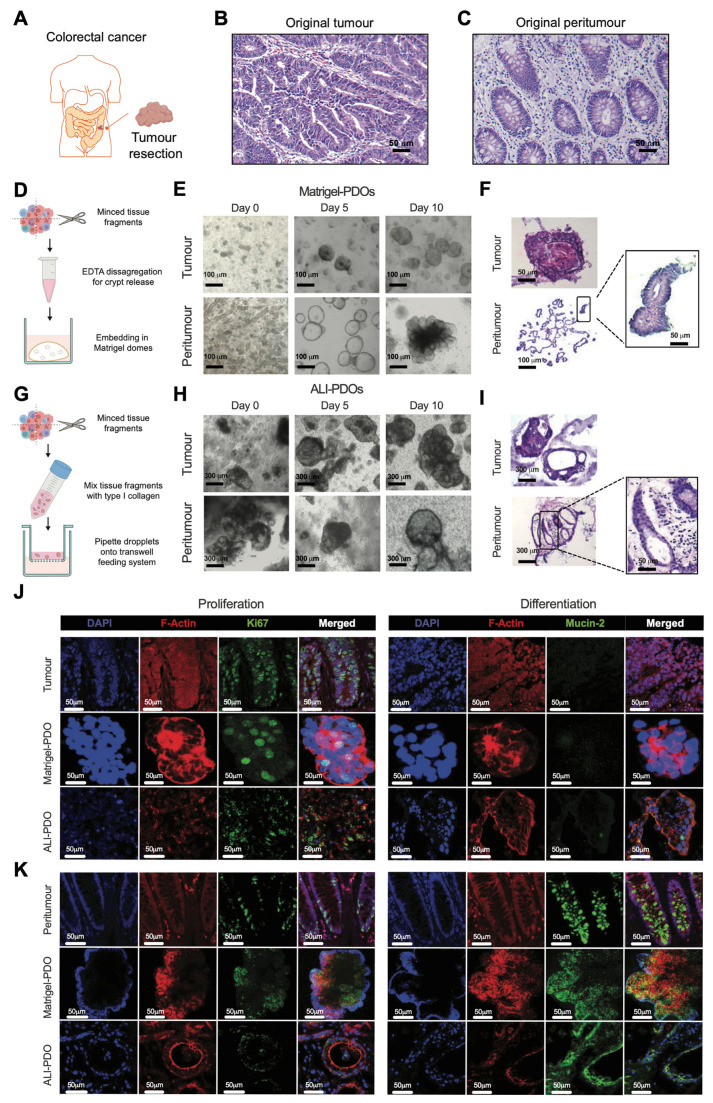
Establishment of Matrigel- and ALI-PDO cultures from colorectal cancer biopsies. Schematic overview of colorectal cancer patient enrolment and tumour resection for organoid generation (**A**) with representative hematoxylin and eosin (H&E) staining of the original tumour tissue (**B**) and matched peritumour tissue (**C**). Scale bars, 50 µm. (**D**) Schematic workflow for the generation of Matrigel-based patient-derived organoids (Matrigel-PDOs). Fresh tissue was mechanically dissociated into fragments, subjected to EDTA-based crypt release, and embedded in Matrigel domes. (**E**) Brightfield images showing the growth of tumour-derived (top) and peritumour-derived (bottom) Matrigel-PDOs at days 0, 5, and 10 of culture. Scale bars, 100 µm. (**F**) H&E staining of sectioned OCT-embedded Matrigel-PDOs derived from both tumour (original magnification: 40×; scale bar: 50 μm) and peritumour tissues (original magnification: 10×; scale bar: 100 μm). On the right panel, a higher magnification (40×; scale bar: 50 μm) of the black frame in peritumour PDOs portraying cell nuclei (purple) at the apical side of the crypt-like domain is shown. (**G**) Schematic workflow for the generation of air–liquid interface patient-derived organoids (ALI-PDOs). Mechanically dissociated tissue fragments were embedded in type I collagen and cultured in a transwell-based air–liquid interface system. (**H**) Bright-field images of ALI-based colon PDOs showing both tumour (top row) and peritumoural (bottom row) samples at various stages. Original magnification: 4×; scale bar: 300 μm. (**I**) H&E images of ALI-PDOs recapitulating the histological architecture of the original tissue. (**J**,**K**) Representative confocal microscopy images in tumour (**J**) and peritumour (**K**). Primary tissues (top row), Matrigel-based PDOs (middle row), and ALI-PDOs (bottom row) stained for DAPI (blue, nuclear marker), Phalloidin (red, F-actin), Ki67 (green, proliferating cells), and Mucin-2 (green differentiation marker for Goblet cells). Original magnification: 20×; scale bar: 50 μm. All images shown are representative of samples derived from patient #5.

**Figure 2 cancers-18-00132-f002:**
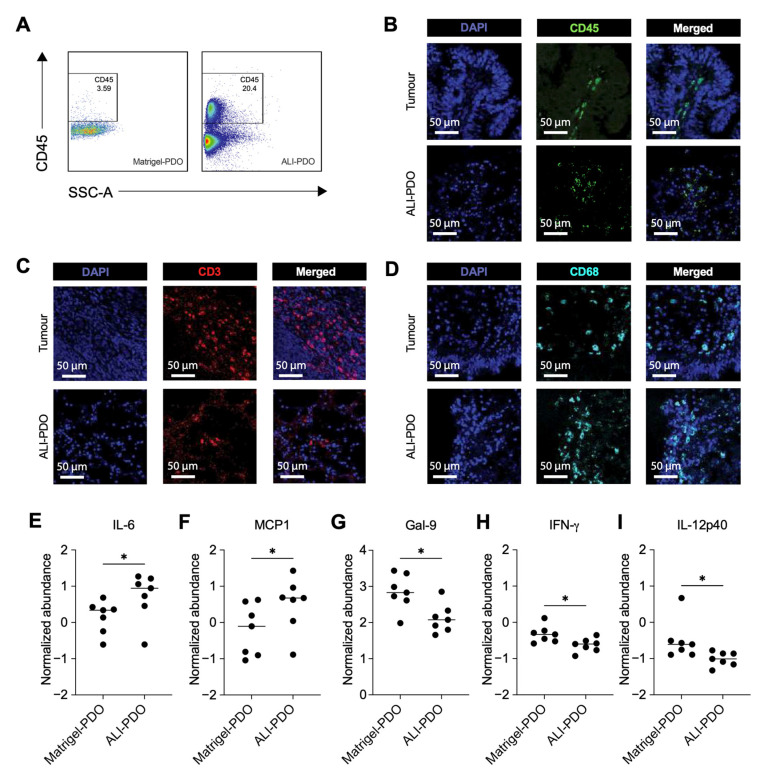
Immune cell composition and soluble immune checkpoint molecules in Matrigel- and ALI-PDOs. (**A**) Flow cytometry dot plots showing the percentage of CD45^+^ immune cells within live cell populations in Matrigel-based PDOs (left) and ALI-PDOs (right). (**B**–**D**) Confocal microscopy images of day-7 ALI-PDOs derived from tumour biopsies stained for CD45 (**B**), CD3 (**C**), and CD68 (**D**), showing the infiltration of immune cells. DAPI (blue) marks the nuclei. Magnification: 20×, Scale bar: 50 μm. (**E**–**I**) Normalized abundance (Z-score) of soluble immune checkpoint ligands and cytokines (IL-6, MCP-1, Galectin-9, IFN-g and IL-12p70, respectively) in the culture supernatants of Matrigel- and ALI-PDOs cultures at day 7, n = 7. Statistical significance was assessed via unpaired Mann–Whitney test (* *p* < 0.05).

**Figure 3 cancers-18-00132-f003:**
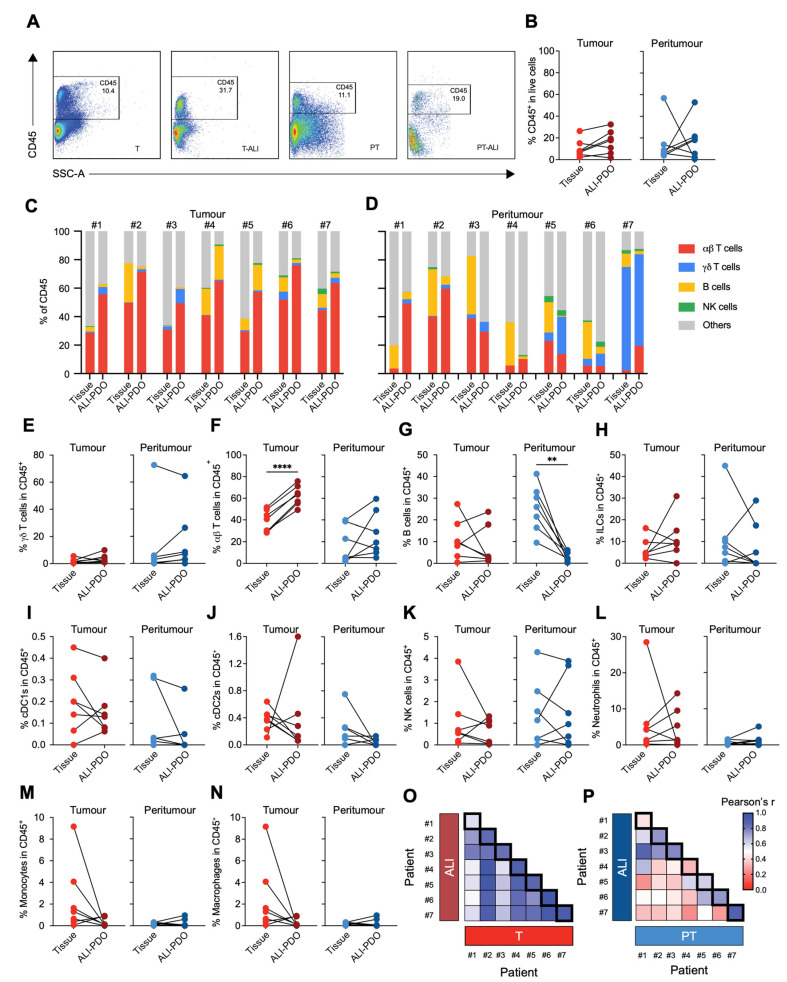
Analysis of immune infiltrates in tumour and peritumoural tissues and corresponding ALI-PDOs. (**A**) Representative flow cytometry dot plots demonstrating CD45^+^ immune cell infiltration in tumour and peritumour tissues alongside their respective ALI-PDOs. (**B**) Percentage of CD45^+^ cells/live cells in matched fresh tissues and ALI-PDOs from 7 CRC patients. (**C**,**D**) Immune cell composition in matched fresh tissues and ALI-PDOs, including αβ T cells, γδ T cells, B cells, NK cells, and other immune cells (Patients #1 to #7). (**E**–**N**) Proportions of γδ T cells (**E**), αβ T cells (**F**), B cells (**G**), innate lymphoid cells (ILCs) (**H**), type 1 conventional dendritic cells (cDC1s) (**I**), type 2 conventional dendritic cells (cDC2s) (**J**), NK cells (**K**), neutrophils (**L**), monocytes (**M**), and macrophages (**N**), n = 7. Statistical significance was evaluated using paired Student’s *t*-tests (** *p* < 0.01, **** *p* < 0.0001). (**O**,**P**) Heatmaps depicting Pearson correlation coefficients between immune profiles in tumour (**O**) and peritumour (**P**) tissues and their corresponding ALI-PDOs.

**Figure 4 cancers-18-00132-f004:**
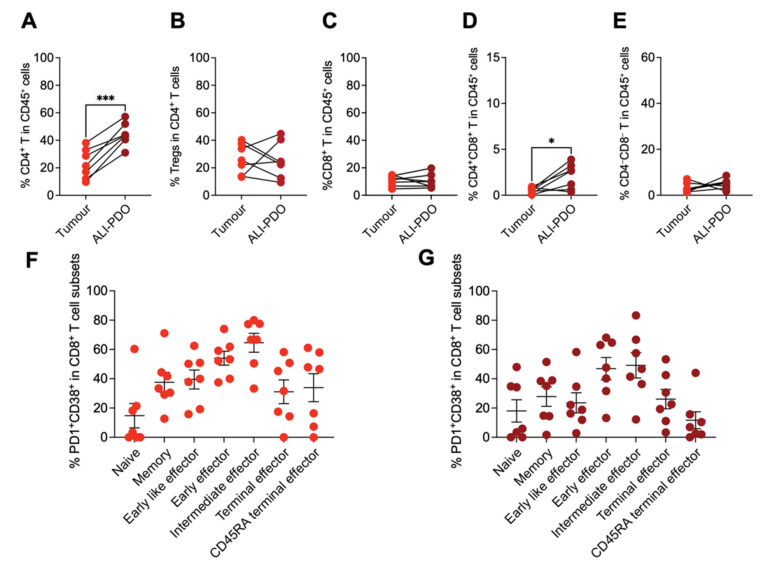
Characterization of tissue-infiltrating T cell subsets in tumour tissues and ALI-PDOs. (**A**–**E**) Characterization of tissue-infiltrating T cell subsets from 7 CRC patients. Quantification of CD4^+^ T cells (**A**), CD4^+^ regulatory T cells (Tregs) (**B**), CD8^+^ T cells (**C**), CD4^+^CD8^+^ T cells (**D**), and CD4^−^CD8^−^ T cells (**E**). (**F**,**G**) Quantification of PD-1^+^CD38^+^ cells within CD8^+^ T cell subsets in tumour tissue (**F**) and ALI-PDOs (**G**). Statistical significance was determined by paired Student’s *t*-test (* *p* < 0.05, *** *p* < 0.001).

**Figure 5 cancers-18-00132-f005:**
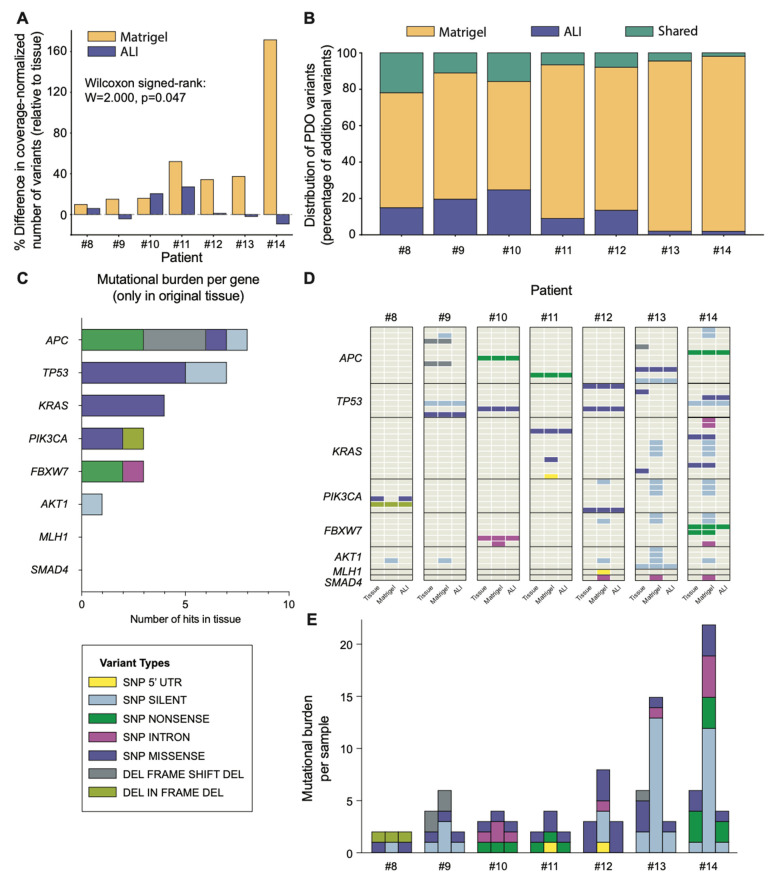
Mutational profiling of primary tumours, Matrigel-, and ALI-PDOs. (**A**) Comparison of the number of variant calls normalized by coverage in WES for original tissue compared to Matrigel and ALI-PDOs (Wilcoxon signed-rank test, *p* = 0.047; Cohen’s d = 0.673). (**B**) Distribution of additional variant calls found in Matrigel-PDOs and ALI-PDOs and shared by both relative to their original tissue. (**C**) Number of mutations found in genes *APC*, *TP53*, *KRAS*, *PIK3CA*, *FBXW7*, *AKT1*, *MLH1* and *SMAD4* throughout the original tissues of this cohort. The figure compares single-nucleotide polymorphisms (SNPs), insertions/deletions (INDELs), and mixed mutations in primary tissues and PDOs. (**D**) Detailed representation of mutations found in either primary tumours, Matrigel- or ALI-PDOs. (**E**) Accumulated number of mutations identified in Matrigel- and ALI-PDOs and their corresponding primary tissues.

**Figure 6 cancers-18-00132-f006:**
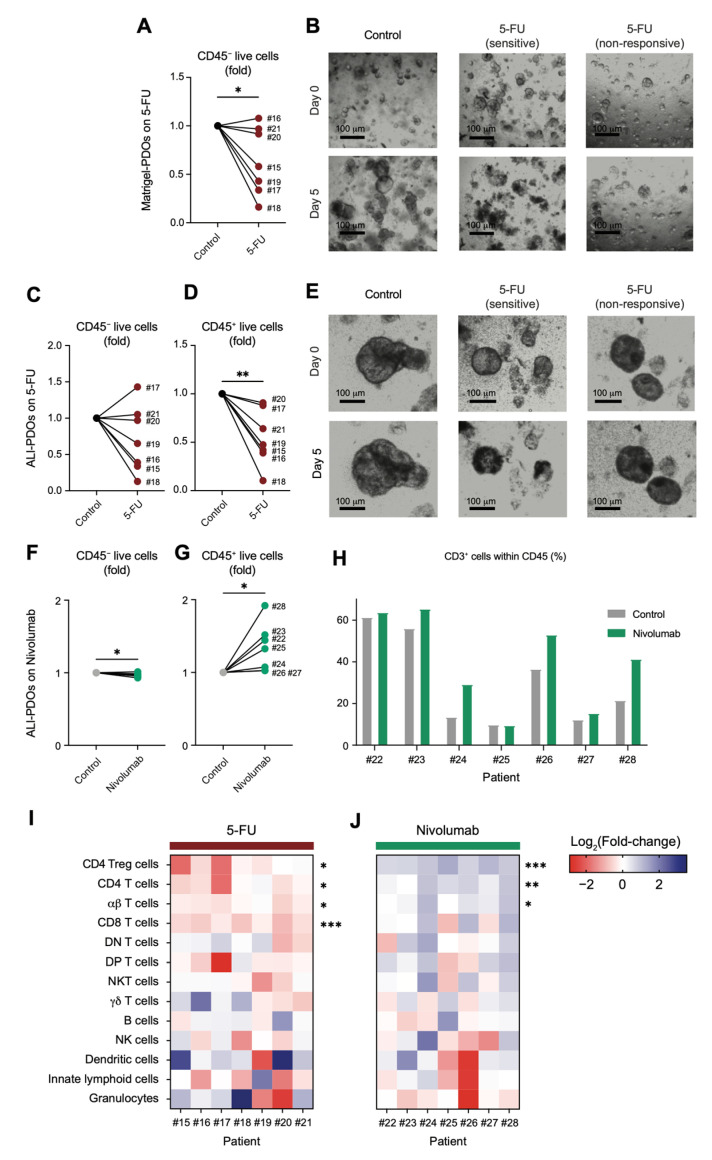
Impact of 5-FU and anti–PD-1 treatment on viability and immune cell composition in Matrigel- and ALI-based PDOs. (**A**) Viability of Matrigel-based PDOs evaluated in CD45^-^ cells following 5 days of 5-FU treatment, n = 7. (**B**) Representative bright-field images illustrating control, 5-FU-sensitive (patient#18), and 5-FU-non-responsive (patient #21) Matrigel-PDOs. Magnification: 10×, scale bar: 100 μm. (**C**) Viability of ALI-PDOs 5 days after 5-FU treatment assessed in CD45^-^ cells, n = 7. (**D**) Viability of CD45^+^ cells after 5 days of 5-FU treatment in ALI-PDOs. (**E**) Representative bright-field images of control, 5-FU-sensitive (patient #18), and 5-FU-nonresponsive (patient #21) ALI-PDOs. Magnification: 10×, scale bar: 100 μm. Viability of ALI-PDOs 5 days after nivolumab treatment assessed in CD45^-^ cells (**F**) and CD45^+^ cells (**G**), n = 7. (**H**) Percentage of CD3^+^ lymphocytes within CD45^+^ in control and nivolumab-treated ALI-PDOs. (**I**,**J**) Heatmaps depicting the fold change induced by 5-FU (**I**) and nivolumab (**J**) treatments on each immune population within tumour-derived ALI-PDOs. Red and blue indicate negative and positive values, respectively. Statistical significance was determined by paired Student’s *t*-test (* *p* < 0.05, ** *p* < 0.01, *** *p* < 0.001).

**Figure 7 cancers-18-00132-f007:**
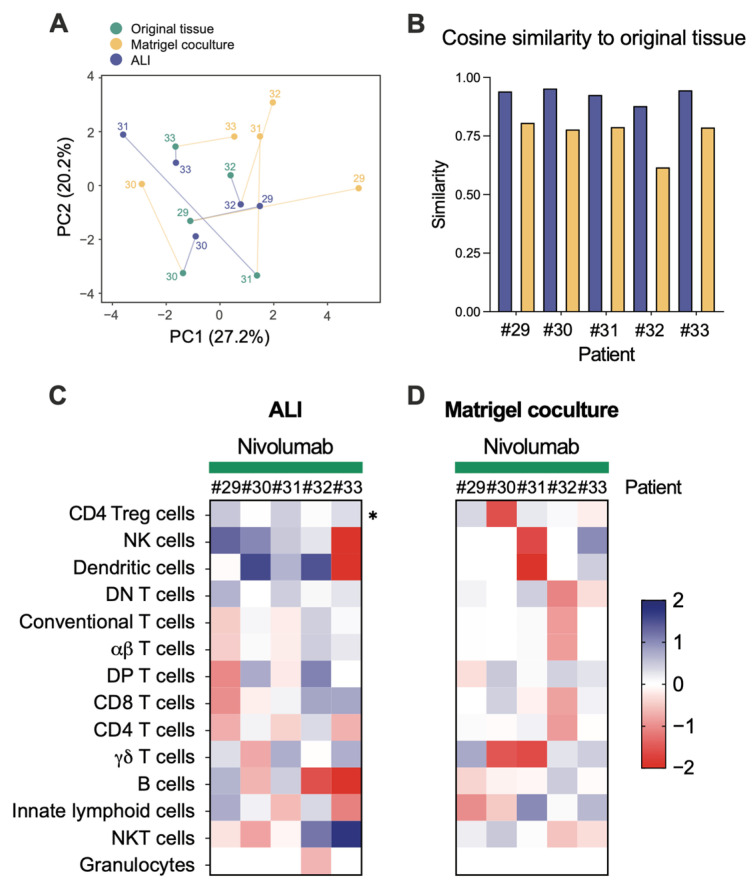
Comparative immune profiling of patient-derived ALI organoids and Matrigel co-cultures with PBMCs. (**A**) Principal component analysis (PCA) illustrating the similarity in immune cell composition between original tumour tissue, ALI organoids and Matrigel co-cultures. (**B**) Cosine similarity indices of each organoid model relative to its matched patient tissue. (**C**) Fold-change in immune composition induced by nivolumab treatment in ALI organoids. (**D**) Fold-change in immune composition induced by nivolumab treatment in Matrigel co-cultures. Statistical significance was assessed using paired Student’s *t*-tests (* *p* < 0.05).

## Data Availability

The Whole Exome Sequencing (WES) data generated and analysed during this study have been deposited in the NCBI Sequence Read Archive (SRA) under accession number [PRJNA1256921]. Further details and documentation are provided within the repository to ensure reproducibility. Any additional information required to reanalyze the data reported in this paper is available from the lead contact upon reasonable request.

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
