# Peer review of "Colorectal Air–Liquid Interface Organoids Preserve Tumour-Immune Architecture and Reveal Local Treg Expansion After PD-1 Blockade"

_cancers, 2025, doi:10.3390/cancers18010132_

Round 1
Reviewer 1 Report
Comments and Suggestions for Authors
Cordoba et al. used two different models of CRC to investigate the effect of immune checkpoint inhibition on T-cells. In the so-called ALI organoid culture system, which is a kind of primary tissue cultivation, the response of the immune system namely the expansion of regulatory T-cells was different to Matrigel-based patient derived organoids co-cultured with PBMCs.
Moreover, the authors comprehensively compared both culture systems with respect to their similarity to the original tumor.
This is an interesting and well-written study. However, it seems to favor the ALI system at times. Here, the authors need to clarify a number of methodological issues and should broaden the discussion to include a critical examination of the ALI cultivation system.
Comments:
I am not comfortable with the idea of calling minced tissue fragments 'organoids'. As I understand it, organoids are self-organizing, 3D structures that can be passaged, cryopreserved and regrown. Thus, experiments can be repeated. This may not be possible with the ALI culture described here.
One major result of the study is the different behavior of immune cells (namely T-cells) in the ALI-culture system compared to co-cultivation of PDOs with PBMCs. This is everything else than surprising, as resident immune cells in the tumor microenvironment are different from PBMCs. Here, it would be of interest to include PBMC co-culture experiments with the ALI culture system. Differentiation between resident immune cells and PMBCs can be achieved with respective staining approaches.
Lines 230ff: How were the results of the LEGENDplex assays normalized? Normalization is pivotal to this kind of assay because different culture systems may contain different cell types, such as tumor cells, fibroblasts, and resident immune cells (e.g., CAFs and TAMs) in ALI cultures versus tumor cells and PBMCs in PDO co-cultures. These cultures may also have different total cell numbers.
Fig. 2 E-K: The question of normalization arises again: Did both systems contain comparable numbers of cells? If not, the difference in ligand/cytokine concentration may simply be due to the different number of cells in each system. To clarify this, the authors should include cell counts.
The discussion favors the ALI culture system. However, the authors should also discuss the disadvantages, such as lack of repeatability, dependency on a fresh tissue supply, and missing information on tumor characterization at the start of the experiment (e.g., mutation status).
Minor comments:
Please use consistent either American or British English (e.g. tumour / tumor)
Line 155: mismatch between “all four stages” and no sample with stage IV was included.
Line 207: Phalloidin is not an antibody. Product # is for an ALEXA 488 stained phalloidin not for rhodamin stained.
Methods section: The authors should provide an illustration depicting the ALI culture system to give readers a better idea of how it works.
Lines 421-424 are confusing. Figure legend and text seem to be mixed up. Or description of Fig 2 E-K is missing in the legend.
Author Response
R: We thank the reviewers for their careful evaluation of our manuscript and for their constructive comments. We have revised the manuscript accordingly and believe that these changes have significantly strengthened both the methodological clarity and the balance of the discussion. Detailed responses to each comment are provided below.
Reviewer #1:
C: Cordoba et al. used two different models of CRC to investigate the effect of immune checkpoint inhibition on T-cells. In the so-called ALI organoid culture system, which is a kind of primary tissue cultivation, the response of the immune system namely the expansion of regulatory T-cells was different to Matrigel-based patient derived organoids co-cultured with PBMCs.
Moreover, the authors comprehensively compared both culture systems with respect to their similarity to the original tumor.
This is an interesting and well-written study. However, it seems to favor the ALI system at times. Here, the authors need to clarify a number of methodological issues and should broaden the discussion to include a critical examination of the ALI cultivation system.
C: We thank the reviewer for their positive assessment of the study and for raising important conceptual and methodological points.
Comments:
C: I am not comfortable with the idea of calling minced tissue fragments 'organoids'. As I understand it, organoids are self-organizing, 3D structures that can be passaged, cryopreserved and regrown. Thus, experiments can be repeated. This may not be possible with the ALI culture described here.
R: We appreciate this conceptual clarification. We agree that freshly minced tissue fragments should not be referred to as organoids. In the revised manuscript, we have clarified our terminology to ensure accuracy and consistency. Specifically, we refer to these early-stage cultures as “tissue fragments” (in Materials and Methods section, lines 195-196), and use the term “organoids” exclusively for structures that have undergone defined culture and exhibit organized epithelial architecture throughout the text and figure legends.
We agree with the reviewer that ALI cultures differ from classical Matrigel-embedded, passagable organoids in their degree of self-organization and long-term expandability. All experiments in this study were performed using early culture time points, as a recognized limitation of ALI systems is the progressive loss of immune infiltrates over time. In the revised manuscript, we have explicitly clarified this distinction in the Discussion, defining ALI-PDOs as short-term, tissue-preserving organotypic cultures rather than expandable stem-cell–derived organoids (Discussion, lines 768-780).
Importantly, the term “ALI-PDO” is used consistently with prior seminal studies (e.g., Neal et al., Cell 2018), and although the departing point may be minced tissue fragments, they still rearrange over time and maintain growth, at least those coming from tumour tissue, as can be seen in the following figure. Please, let us know if the reviewer considers this panel should be included as supporting data.
Figure 1 for Reviewer 1. Organoid area was calculated using Image J software of seven randomly selected Matrigel- and ALI-PDOs derived from tumour and peritumoral tissues at culture days 5 and 10. Graphs show ± mean SEM. Statistical significance was evaluated using two-way ANOVA (*p < 0.05, ****p<0.0001).
One major result of the study is the different behavior of immune cells (namely T-cells) in the ALI-culture system compared to co-cultivation of PDOs with PBMCs. This is everything else than surprising, as resident immune cells in the tumor microenvironment are different from PBMCs. Here, it would be of interest to include PBMC co-culture experiments with the ALI culture system. Differentiation between resident immune cells and PMBCs can be achieved with respective staining approaches.
We agree that the divergent behaviour of immune cells between ALI cultures and PBMC co-cultures is not unexpected, given the fundamentally different immune compartments involved. Our intention was not to imply equivalence between resident tumour-infiltrating lymphocytes (TILs) and circulating PBMCs, but rather to highlight how these biological differences might translate into distinct responses to immune checkpoint blockade and allow for observations more adjusted to the TME that might be overseen when using PBMC cocultures. We have now emphasized this rationale more explicitly in the Discussion, lines 678-689.
Regarding the suggestion to combine PBMCs with ALI cultures, we agree that it could represent an interesting experimental extension. Such an approach could provide insight into whether tumour-resident immune cells are capable of conditioning or instructing incoming peripheral immune populations within a tumour-like microenvironment. However, this strategy would introduce additional variables that fall outside the scope of the present study, which was to preserve and interrogate endogenous tumour–immune interactions without the confounding influence of exogenous immune cells. Moreover, PBMC-ALI co-cultures would still face the challenge of defining physiologically relevant immune-to-tumour cell ratios. Inadequate or excessive numbers of PBMCs could again obscure the behaviour of resident immune populations, potentially limiting the interpretability of such experiments. For these reasons, while PBMC-ALI co-cultures represent a promising avenue for future investigation, they were beyond the scope of the current work.
C: Lines 230ff: How were the results of the LEGENDplex assays normalized? Normalization is pivotal to this kind of assay because different culture systems may contain different cell types, such as tumor cells, fibroblasts, and resident immune cells (e.g., CAFs and TAMs) in ALI cultures versus tumor cells and PBMCs in PDO co-cultures. These cultures may also have different total cell numbers.
Fig. 2 E-K: The question of normalization arises again: Did both systems contain comparable numbers of cells? If not, the difference in ligand/cytokine concentration may simply be due to the different number of cells in each system. To clarify this, the authors should include cell counts.
R: We thank the reviewer for this important point and fully agree that normalization is a critical consideration when interpreting supernatant-based immunomodulator measurements, particularly in culture systems with distinct cellular compositions. Our initial intention was for these data to provide a qualitative view of the capacity of ALI-PDOs to generate an immunomodulatory milieu, reflecting their preservation of multiple tumour-associated cell types (tumour epithelium, stromal cells, and resident immune populations), rather than to quantify per-cell secretion rates.
We agree, however, that differences in total cell numbers between ALI-PDOs and Matrigel-PDO could contribute to differences in absolute cytokine and ligand concentrations. Unfortunately, precise normalization to cell number was not performed in this setting and is not possible at this stage. We carefully considered several alternative normalization strategies, including soluble protein release (e.g., LDH), tissue- or colon-specific markers, DNA content, and total protein concentration. However, each of these approaches introduces substantial bias in this context: LDH and related markers preferentially reflect cell death, which may differ between culture systems; DNA-based normalization similarly does favour the quantification of non-viable cells; and total protein measurements are confounded by the distinct extracellular matrices used (Matrigel versus collagen I).
To address these limitations, we have implemented a complementary normalization strategy based on log-transformation followed by within-sample Z-score scaling across immunomodulators. IL-2, which was supplemented in the ALI model, was excluded from the normalization algorithm. This approach allows comparison of relative immunomodulatory programs within each culture system, independent of global secretion magnitude or total cell number. Importantly, this approach still reveals relevant differences between ALI-PDOs and Matrigel-PDOs—particularly the relative enrichment of inflammatory mediators in ALI-PDOs—were preserved using this normalization method. This may reflect the capacity of ALI-PDOs for preserving native immune infiltrates, especially those of the myeloid lineage.
Something to bear in mind, is that, while this approach allows for in-sample normalization to visualize relative immunomodulator programs, it does not provide a per-cell production and that some immunomodulators could be biased. Namely, this approach can highlight differences of those that may be more relevant, while obscuring differences in some other immunomodulators. We have now clarified this rationale in the Methods (lines 254-257) and Results sections (lines 463-471) and discuss its implications in the Discussion (lines 713-721).
C: The discussion favors the ALI culture system. However, the authors should also discuss the disadvantages, such as lack of repeatability, dependency on a fresh tissue supply, and missing information on tumor characterization at the start of the experiment (e.g., mutation status).
R: We fully agree with the reviewer that a balanced discussion of the ALI culture system is essential. Our intention was not to uncritically favour the ALI platform, but rather to evaluate its strengths in the context of specific experimental questions. Accordingly, we have now added a dedicated paragraph in the Discussion (lines 768–780) explicitly addressing the limitations of ALI-PDOs, including their limited repeatability, reliance on freshly resected tissue, and constraints on comprehensive tumour characterization at the start of the experiment (e.g., mutation status). We believe this addition provides a more balanced and transparent assessment of the ALI system and clarifies its role as a complementary, rather than superior, model to conventional Matrigel-based PDOs.
Minor comments:
C: Please use consistent either American or British English (e.g. tumour / tumor)
R: We thank the reviewer for this observation. We have improved the manuscript accordingly.
C: Line 155: mismatch between “all four stages” and no sample with stage IV was included.
R: As indicated in supplementary table 1, no samples were obtained from stage IV CCR patients. Thus, we have now corrected the maintext.
C: Line 207: Phalloidin is not an antibody. Product # is for an ALEXA 488 stained phalloidin not for rhodamin stained.
R: Thanks for the observation. It is now corrected as reagents.
C: Methods section: The authors should provide an illustration depicting the ALI culture system to give readers a better idea of how it works.
R: Great idea, also proposed by Reviewer 2. Now included in Figure 1.
C: Lines 421-424 are confusing. Figure legend and text seem to be mixed up. Or description of Fig 2 E-K is missing in the legend.
R: Thanks for bringing it up. It has now been corrected. The error came from the formatting process.
Reviewer 2 Report
Comments and Suggestions for Authors
Overall comments
The manuscript by Córdoba et al. presents a systematic study of colon cancer organoids generated using an air–liquid interface approach. The work is carefully performed, with a range of appropriate endpoints assessed. A notable advantage of this organoid system is its ability to preserve an active immune cell component. While the report offers few novel findings, its thorough documentation and validation will be highly valuable to this group and others in advancing new strategies for colon cancer therapy, particularly immune-based approaches.
Specific comments
- It would be helpful to include a schematic showing the procedure for making the two different types of organoids in Figure 1.
- It would be helpful to show and H&E stain of the original tumor in Fig 1.
- The description of Fig 6 in the text didn’t seem to match what was shown in the figure. The authors should double check this section of the manuscript. Overall, it was difficult to understand the conclusions from the 5FU experiments.
- One outcome of nivolumab treatment in ALI-PDOs was the expansion of Tregs. This observation is significant because it has also been reported in patients and likely contributes to limiting the effectiveness of immune checkpoint inhibitors. Although this finding is mentioned in the abstract, its importance does not become clear until later in the paper. Emphasizing this point more strongly in the abstract and Introduction is advised
Author Response
Reviewer #2:
C: The manuscript by Córdoba et al. presents a systematic study of colon cancer organoids generated using an air–liquid interface approach. The work is carefully performed, with a range of appropriate endpoints assessed. A notable advantage of this organoid system is its ability to preserve an active immune cell component. While the report offers few novel findings, its thorough documentation and validation will be highly valuable to this group and others in advancing new strategies for colon cancer therapy, particularly immune-based approaches.
R: We thank the reviewer for their positive evaluation of the study and for recognizing the value of this platform for immune-based CRC research.
Specific comments
- It would be helpful to include a schematic showing the procedure for making the two different types of organoids in Figure 1.
R: We agree and have added two layouts in Figure 1 illustrating both Matrigel-PDO and ALI-PDO generation workflows (Panels D and G).
- It would be helpful to show and H&E stain of the original tumor in Fig 1.
R: Thanks for the insight, it has now been included as panels B and C. Lines 362-409 describe the new panels.
- The description of Fig 6 in the text didn’t seem to match what was shown in the figure. The authors should double check this section of the manuscript. Overall, it was difficult to understand the conclusions from the 5FU experiments.
R: We thank the reviewer for this observation. We carefully re-examined the description of Figure 6 and confirmed that the data and referenced panels were correct. However, we agree that the wording of the corresponding Results text did not sufficiently guide the reader through the figure, which may have caused confusion regarding the interpretation of the 5-FU experiments.
To address this, we have revised and clarified the text describing Figure 6 to ensure a more accurate and transparent correspondence between the narrative and the displayed data (lines 603-619). In addition, we have reworded the Results section to more clearly state the main conclusions of the 5-FU experiments, particularly with respect to the heterogeneity of responses observed across models and patients. We believe these changes improve the clarity and readability of this section without altering the underlying results or conclusions.
- One outcome of nivolumab treatment in ALI-PDOs was the expansion of Tregs. This observation is significant because it has also been reported in patients and likely contributes to limiting the effectiveness of immune checkpoint inhibitors. Although this finding is mentioned in the abstract, its importance does not become clear until later in the paper. Emphasizing this point more strongly in the abstract and Introduction is advised
R: We agree with the reviewer that the expansion of regulatory T cells following PD-1 blockade represents a key finding of the study with important clinical implications. While this observation was described in the Results and Discussion, we have now strengthened its emphasis earlier in the manuscript. Specifically, we have revised the Abstract to explicitly highlight Treg expansion as a central outcome of nivolumab treatment and its potential role in limiting immune checkpoint efficacy. In addition, we have expanded the Introduction (lines 170-179) to frame Treg expansion as a clinically relevant mechanism of immune resistance that motivated our experimental approach. We thank the reviewer for this suggestion, which helps to strengthen the significance of this finding from the outset of the manuscript.
Please, find that we have made several minor corrections that had been brought from previous versions of the manuscript. Some author affiliations and contributions have been adapted. Some text from the methods section has been removed as it is not relevant to the current manuscript. Results (lines 591-596) that help describe the figure and do not modify the conclusions of the manuscript have been included.